### Is the summer aerosol over the Arctic controlled by regional 1

### atmospheric circulation or ice conditions? Trends and Future 2

## **Implications** 3

- Caroline Leck<sup>1</sup>, Jost Heintzenberg<sup>2</sup>, Tiina Nygård<sup>3</sup>, Tuomas Naakka<sup>3</sup> 4
- 5 <sup>1</sup>Department of Meteorology and Bert Bolin Centre for Climate Research, Stockholm University, Stockholm, 106
- 91, Sweden 6
- 7 <sup>2</sup>Leibniz Institute for Tropospheric Research, Leipzig, 04318, Germany
- 8 9 <sup>3</sup>Finnish Meteorological Institute, PB 503, Helsinki, 00101, Finland

- 10 Correspondence to: Caroline Leck (lina@misu.su.se)
- 11 **Abstract.** Based on aerosol particle number size distributions measured ≥ 85° N on I/B Oden covering the summers 12 of 1991, 1996, 2001, 2008, and 2018, regional atmospheric circulation regimes (nodes) delineated with self-13 organizing maps (SOMs) were investigated as potential controllers of Arctic aerosol sources. The three most 14 prominent nodes were not connected to regional source-related differences and did not vary systematically 15 throughout the study period. Instead, the seasonal course of sea ice melt and freeze-up appeared to affect the shape of the aerosol size distributions significantly. High sub-Aitken concentrations occurred during the "freeze-up", 16 17 most commonly associated with the low wind, restricted sea ice movement, and effective radiative cooling. The high concentrations of newly formed particles measured during "freeze-up" were interpreted as deriving from frost 18 19 flower formation. With the data on ice and atmospheric conditions and their seasonal course, the study was 20 extended to cover all years from 1991 to 2023 to enable speculations about changing aerosol source conditions in 21 the warming Arctic climate. Over the 33 years of the study, the significant increases in sea and air temperatures 22 nearly doubled the favorable ice conditions for new particle formation ≥ 85° N, lengthening both "melt" and 23 "freeze-up" parts of the illuminated Arctic by more than a week. Whereas the sum effect of counteracting 24 processes during the ice melt season on the airborne biogenic Arctic aerosol in a warming climate is unclear, the 25 net effect of the changing the freeze-up of sea ice is expected to enhance the biogenic Arctic aerosol in late

## 1 Introduction

summer/autumn.

37 Over the past few decades, climate change has had a profound impact on the Arctic, affecting it more than any other region on Earth. Averaged over the years 1979-2012, the temperature north of 66.5° N has risen almost four times faster than the global average in recent decades (Rantanen et al., 2022). One of the most noticeable consequences is the alarming reduction in the extent and mass of sea ice, which occurs in all seasons but is most dramatic in late summer, when the sea ice extent reaches its annual minimum (Meier et al., 2014). The minimal ice cover in late summer follows the gradual creation of melt ponds and reduction of dry ice caused by solar energy input and rising air temperatures, which commence as the sun rises above the horizon in March. This period is referred to as "melt." During the start of new ice formation, called "freeze-up," the first ice layer forms when the sea temperature is close to its freezing point, dropping below ≈ -1.8° C. This first layer of greased ice rapidly solidifies into thin sheets, thickening through rafting and ridging processes until it is fully frozen (Comiso, 2010).

During the freezing process, saline brine gets trapped within ice crystals, creating a brine-wetted surface on the new ice. Highly saline centimeter-scale frost flowers will also form when the brine migrates upward or is expelled from the sea surface under the new ice, which is typical during high-pressure atmospheric systems with calm winds and cold temperatures (< -8°C) (Galley et al., 2015;Perovich and Richter-Menge, 1994).

Low-altitude liquid clouds are particularly important in Arctic climate change. By influencing the surface energy budget, these clouds can partially offset regional warming. These clouds in the summer high Arctic north of 80° contain fewer but larger droplets than clouds in other regions (Mauritsen et al., 2011). Coupled with the semi-permanent ice cover, even small changes in either can significantly influence heat transfer to the ice and its melting process. As such, the regional aerosol and its sources over the pack ice potentially play a significant role in regulating the surface energy budget through aerosol-cloud interactions. Ceteris paribus, if more aerosol becomes available for water uptake, clouds may form with numerous smaller droplets. This increases their sunlight reflection, leading to surface cooling (Twomey, 1974).

The air mass analyses of Heintzenberg et al. (2015) showed that the summer aerosol over the pack ice has different potential source regions both within the pack ice itself and along its edge at the Marginal Ice Zone (MIZ). The Arctic's synoptic-scale atmospheric circulation exhibits strong seasonal patterns along with notable interannual variability. This is evident in large-scale seasonal shifts in the location and strength of cyclones, their primary pathways, and maxima of anticyclones. To generalize, cyclonic activity in winter and spring is mainly confined to the eastern Arctic, with local maxima near Svalbard, and the northern part of Novaya Zemlya, and in winter, east of the North Pole. These systems are migratory, primarily entering from the North Atlantic and Barents Sea. Anticyclone maxima occur in the Canada Basin, in the sector encompassing the Siberian, Chukchi, and Beaufort Seas, extending up to 85° N. The summer circulation pattern has attracted most research attention among the seasons due to its close timing with the September sea-ice minimum. Unlike winter and spring, the Canada Basin experiences its highest cyclone activity during summer, driven by land-ocean temperature differences and the arrival of mid-latitude cyclones. Systems mainly migrate into this region from along the Siberian coast, resulting in persistent low-pressure systems. The autumn season serves as a transitional phase, exhibiting a combination of summer and winter circulation patterns (see Ding et al., 2017;Lee and Kim, 2019;Serreze and Barrett, 2008 for more details).

Another conclusion from Heintzenberg et al. (2015) indicated that ice conditions with 10% to 30% open water a few days before the air mass arrived at the observation site promoted the presence of aerosol sources. Previous findings over the Arctic pack ice during "melt" showed that local emissions of nascent organic sea spray aerosols, from the upper ocean's microbial community, can alter particle concentrations or composition (e.g., organic fragments or coatings on salt particles: Leck et al., 2002;Leck and Bigg, 2005a;Leck and Svensson, 2015). Orellana et al. (2011) confirmed that organic material in near-surface aerosols acts like marine polymer gels<sup>1</sup>, originating from the surface microlayer (SML) on open leads<sup>2</sup>. It should be noted that the contribution of inorganic salts from sea spray is observed to have a negligible impact on the number population of cloud-active aerosols over the pack ice in summer (Leck et al., 2002). Moreover, Bowman and Deming (2010) found that frost flowers formed during "freeze-up" have higher levels of bacteria and extracellular polymer gels than brine, young ice, or lead water. Their

<sup>&</sup>lt;sup>1</sup> Marine polymer gels are exudates from phytoplankton, ice algae, and bacteria consisting of Ca<sup>2+/</sup>Mg<sup>2+</sup> cross-linked polysaccharides that bind together small particulates and organic molecules such as amino acids, peptides, proteins, and lipids (Orellana et.al., 2021).

<sup>&</sup>lt;sup>2</sup> Open leads are openings of sea water in pack ice and characteristically form long, narrow channels, 1-100m wide and up to kilometers long.

findings suggest that frost flowers allow SML and young sea ice to interact chemically with the atmosphere, potentially serving as a source of polymer gels. Another piece of evidence supporting the connection between marine life and new particle formation is the reaction of iodic acid and sulfuric acid (Baccarini et al., 2020; Beck et al., 2021). Iodic acid is believed to form from atmospheric photooxidation of iodine, which is mainly produced by microalgae beneath sea ice and released through brine channels and frost flowers (Saiz-Lopez et al., 2015), or possibly through abiotic processes from pack ice, especially during "freeze-up". During advection over the pack ice, sulfuric acid forms through the photooxidation of gas-phase dimethyl sulfide (DMS), which is primarily influenced by the conditions of the marine microbial food web in the open ocean and in the wake of the receding ice at the MIZ, rather than locally within the pack ice (Karl et al., 2007; Kerminen and Leck, 2001; Leck and Persson, 1996a,b).

Consequently, in the present study, the question of whether regional-scale atmospheric circulation patterns or the temporal development of the "melt"/"freeze-up" season plays a key role in summer aerosol sources over the inner Arctic is pursued. Here, we present a synopsis of all aerosol number size distributions accumulated during five I/B *Oden* expeditions, 1991-2018, to the inner Arctic north 85-90° N (Leck et al., 1996; Leck et al., 2001; Leck et al., 2004; Tjernström et al., 2014; Leck et al., 2019). The largest number of observations occurred while I/B *Oden* was moored to an ice floe in the inner pack ice area between 85-90° N, drifting passively. This period marked the transition from the biologically most active summer "melt" to the autumn "freeze-up", roughly from mid-August to mid-September.

As a first step, to provide context for the relatively short observation periods from each of the five expeditions, the self-organizing maps (SOMs) classification method was employed to identify Arctic circulation patterns during the summer-to-autumn months of August and September. The SOM method uses unsupervised learning to identify generalized patterns in data and, consequently, clusters a large volume of synoptic pressure fields based on similar large-scale circulation distributions. Each timestep of the input data will belong to one of the resulting circulation regimes called nodes. The SOM method will also provide circulation distributions or regimes (nodes) of, e.g., horizontal moisture transport, total cloud water, radiation, evaporation, and surface temperature. The nodes were subsequently linked to the number size distributions of all aerosol number size distributions accumulated during the past five I/B *Oden* expeditions covering the years 1991-2018.

Connecting air mass analyses with sea ice cover revealed that broken ice conditions favor aerosol sources over the inner Arctic (Heintzenberg et al., 2015). The second focus of the present study was stimulated by these findings, which employed the detailed seasonal evolution of sea ice to understand its impact on aerosol sources and factors affecting the shape of their size distributions over the summertime Arctic pack ice area. The daily ice maps were used to analyze sea ice conditions during the "melt" and "freeze-up" periods. This analysis was conducted for August and September each year from 1991 to 2018. The data from the ice maps were then connected to all aerosol number size distributions compiled over the past five I/B *Oden* expeditions.

In the final part of the study, SOMs covering all summers from 1991 to 2018, along with ice maps, sea surface temperatures, and atmospheric temperature data from ERA5 (the fifth generation of the European Centre for Medium-Range Weather Forecasts, ECMWF), were integrated up to 2023 across two geographic regions. The inner Arctic pack ice region, located at latitudes greater than or equal to 85° N, was compared with the marginal ice region between 78° N and 82° N. With marginal ice and advanced summer melt, the latter region can reference today's conditions that may govern the inner Arctic within a few decades, as indicated in Fig. 1 of Wassman and

Reigstad (2011). This extensive dataset explores potential long-term trends and future implications of atmospheric

and ice conditions for Arctic aerosol sources in summer.

## 2 Methods

# 2.1 Self-Organizing Maps and Surface Air and Ocean Temperatures

The European Centre for Medium-Range Weather Forecasts (ECMWF) Reanalysis v5 (ERA5) mean sea level pressure (MSLP) fields were clustered using the Self-Organizing Maps (SOM) method to identify the Arctic's main large-scale atmospheric circulation regimes. The SOM method, developed by Kohonen (2001), is an unsupervised learning method, i.e., a machine-learning approach, to identify generalized patterns in data. The method has previously proven valuable in atmospheric applications (Nygård et al., 2019;Thomas et al., 2021),

providing physically meaningful composites of field patterns.

The MSLP data were collected at 12-hour intervals, covering days from August to September 1991–2018, and served as input for the SOM analyses. These input data were chosen because they aligned with the periods when shipboard measurements were most frequently available.

As a first step, the MSLP data were re-gridded to an equal-area grid. Then, the SOM algorithm created an initial SOM array containing six nodes with random reference vectors of an equal dimension as the input MSLP data. After that, each input data vector was compared with the reference vectors, and the reference vectors most similar to the input data vector were adjusted towards the input data vector. This was repeated until the reference vectors converged. Finally, the SOM algorithm provided an organized SOM array of MSLP patterns, having the most similar nodes (i.e., circulation regimes) next to each other. However, in this paper, composites of the MSLP fields associated with each node were presented, not the output reference vectors of the SOM analysis. For a more detailed description of the SOM method, please see Kohonen (2001) and Hewitson and Crane (2002).

The choice of the SOM output array size is always, to some extent, subjective (Alexander et al., 2010). The results for a 2 x 3 array and a 3 x 4 array were compared, and it was concluded that the 2 x 3 array should be proceeded with. These six nodes in the 2 x 3 array can adequately represent the range of large circulation patterns in the Arctic, so that their variation is captured in the variation of the circulation patterns of the nodes in enough detail for the aims of the study. It is also beneficial for only a few nodes to be present when our aerosol observational data was later associated with the circulation regimes; this will ensure that sufficient observational data is available to be associated with each node to provide statistically representative results.

The European Centre for Medium-Range Weather Forecasts (ECMWF) Reanalysis v5 (Hersbach et al., 2020; Hersbach, 2023, (last accessed 2024-10-31)) mean sea level pressure (MSLP) fields were clustered using the SOM method to identify the Arctic's main large-scale atmospheric circulation regimes. ERA5 is a state-of-the-art global atmospheric reanalysis that applies a four-dimensional variational data assimilation method to assimilate various observations. Uncertainties in the representation of MSLP fields by ERA5 are assumed to be minor, as in global reanalyses in general (Nygård et al., 2021). However, uncertainties in ERA5 radiation, especially cloud variables, may be considerably more significant (Nygård et al., 2021). In this study, the means of variables, including 10-meter wind vectors, temperatures (at 2 meters and 850 hPa), vertically integrated moisture vectors, total cloud water, net longwave (LW) radiation, and surface temperature in °C were calculated separately for each of the six MSLP circulation regimes. In addition, an extended period of two meter-air temperatures from the ERA5 analyses for August and September of 1991-2023 was also utilized. Arctic-wide average temperature values north of 85° N

were calculated  $\pm 6h$  about each time step of the SOMs to interpret SOMs, sea ice, and aerosol data. Half-day mean (06-18 and 18-06) 2m temperature (T2m) and Sea Surface Temperature (SST) values were calculated from hourly ERA5 data. Areal-mean time series were constructed using half-day mean values for two distinct areas: the entire inner Arctic pack ice region north of 85° N and the marginal ice zone within the 78° N – 82° N latitude

band.
Only ocean areas where the land fraction is less than 0.1 were included in the calculation. T2m was calculated

in the ERA5 assimilation cycle. Still, since the atmospheric model used to produce the ERA5 reanalysis does not include ocean model SSTs, these are provided as input for assimilation. Two datasets have been used to provide

SST values for ERA5. Before September 2007, SST for ERA5 was supplied by the HadISST2 dataset (Titchner

and Rayner, 2014), and from September 2007 onward, it was provided by the OSTIA dataset (Good, 2022).

## 2.2 Aerosol measuring systems and platforms

Number-size distributions of aerosol particles have been measured on I/B *Oden* on all five Arctic expeditions.

From 1991 through 2008, the same type of differential mobility analyzer was deployed, albeit with varying upper

and lower size limits and the number of steps in particle diameter. Relative humidities in the instruments were

below 20%. Direct ship contamination was prevented using a pollution controller connected to the sampling

manifold, consisting of a TSI-3025 UCPC linked to the system described by Ogren and Heintzenberg (1990).

Aditionally, safe wind sectors were identified by releasing smoke from the ship while adjusting wind speed and

direction (Leck et al., 1996). If the wind was within ±70° of the bow and above 2 m s<sup>-1</sup>, no ship pollution reached

the sample inlets. In 2001 and 2008, a third criterion was added: excluding data when one-minute toluene levels

exceeded 75% of their running mean. To maximize pollution-free sampling time, the manifold was kept facing

upwind, requiring a "harbor" in the ice to moor the ship and allow rotation with changing wind directions. Details

of the respective instrument setups and measures to minimize the risk of contamination from the platform are

described in Covert et al., (1996), Leck et al., (2001), and Heintzenberg and Leck (2012).

In 2018, a new type of aerosol spectrometer was added to a scanning differential mobility analyzer that extended the size range down to one nanometer (Baccarini et al., 2020; Karlsson and Zieger, 2020). Whenever the measured data did not cover this set completely the values at the respective interpolation diameters were flagged as 'missing'. The size distributions from the five expeditions were harmonized before the fitting procedure by linear interpolation of the measured data on a common set of 50 diameters from 3.37 to 900 nanometers with logarithmically equal spacing. Whenever the measured data did not cover this set completely, the values at the respective interpolation diameters were flagged as 'missing'.

Previous studies (Covert et al., 1996; Heintzenberg and Leck, 2012) had shown that the surface aerosol in the inner Arctic exhibited number size distributions as a combination of several modes. These included nucleation modes below 10 nanometer, ultrafine particle modes below 25 nm, Aitken modes with a maximum between 25 and 60-80 nm, and accumulation modes above 60-80 nm in diameter. The lognormal fitting was accomplished with an algorithm written in FORTRAN that follows the procedure described by Whitby and McMurry (1997). The multimodal character of the sub-micrometer aerosol size distribution is well established in the summer Arctic. Thus, with two to six lognormal modes, a given number size distribution is approximated over the total diameter range 1-1000 nm by randomly varying positions, logarithmic standard deviations, and total number concentrations of the modes. When an optimal fit is arrived at, the number of modes is reduced as much as possible while keeping the average difference between model and measurement below a given upper limit. Absolute concentrations

should be maintained within 50% of the measurements, while relative differences between model and measurements are maintained within 25%. The latter condition secures a good simulation of the wings of the particle size distribution. Statistics of the five expeditions in Table A1 show that on average the quality of the lognormal approximations are substantially better than given by the constraints of the algorithm and less than 10% of the could not be approximated within the set limits. Only aerosol data  $\geq 85^{\circ}$  N were utilized for our focus on the inner Arctic. The exact periods and numbers of used hourly aerosol data are collected in Table 1. Aerosol data coverage in relation to "melt" and "freeze-up" is shown in Fig. A1 in the Appendix.

**Table 1:** Start and end date of hourly I/B *Oden* aerosol data utilized in this study in 1991, 1996, 200, 2008, and 2018, and the number of utilized hourly averaged distributions (Scans)  $\geq$  85° N after screening for possible ship pollution (total 2476). Also shown are the aerosol upper and lower size detection limits of the instruments used in the different expedition years.

| Year | Start date | End date     | Lower size<br>limit (nm) | Upper size<br>limit (nm) | Scans |
|------|------------|--------------|--------------------------|--------------------------|-------|
| 1991 | 23 August  | 20 September | 3                        | 500                      | 560   |
| 1996 | 1 August   | 9 September  | 5                        | 600                      | 715   |
| 2001 | 1 August   | 24 August    | 3                        | 900                      | 503   |
| 2008 | 10 August  | 3 September  | 3                        | 800                      | 411   |
| 2018 | 15 August  | 16 September | 2.1                      | 921                      | 287   |

## 2.3 Air-mass back trajectories

Hourly five-day air-mass back trajectories were calculated arriving at I/B *Oden*. They cover each hour of the utilized size distribution data. The trajectories were based on the meteorological fields kindly made available by the US National Weather Service's National Center for Environmental Prediction (NCEP). Before 2005, the trajectories were based on NCEP/NCAR reanalyzed meteorological fields with  $2.5^{\circ} \times 2.5^{\circ}$  resolution (https://www.ready.noaa.gov/gbl\_reanalysis.php, last accessed 2023-09-08). After 2005, the calculations were based on the Global Data Assimilation System (GDAS1). In horizontal grids of  $1^{\circ} \times 1^{\circ}$  resolution, meteorological parameters are stored every three hours with a vertical grid spacing of 23 pressure surfaces between 1000 and 20 hPa. All higher layers (with the exception of the top layer) are separated by 50 hPa, (Kanamitsu, 1989). The HYSPLIT-model for trajectory calculation, (Stein et al., 2015), analyzes the meteorological inputs to determine the appropriate internal vertical model resolution so that there are sufficient levels to interpolate all the meteorological input without skipping data due to insufficient vertical resolution.

For the present study, the trajectory ensemble option of HYSPLIT was used. It starts multiple trajectories from a given starting location to estimate the uncertainty associated with the center point trajectory. Each member of the trajectory ensemble is calculated by offsetting the meteorological data by a fixed grid factor (one meteorological grid point in the horizontal and 0.01 sigma units in the vertical). This results in 27 members for all possible offsets in X, Y, and Z (https://www.ready.noaa.gov/hypub-bin/trajtype.pl, last accessed 2024-02-22).

The summer Arctic boundary layer is typically well-mixed and shallow (approximately 100 - 400 m), capped by a temperature inversion. At times, the inversion can be strong, especially when warmer air is advected from lower latitudes while the free troposphere remains stably stratified (Vüllers et al., 2021). An arrival height of 300

meters was chosen to ensure an optimal ensemble configuration, such that the receptor point is within the well-mixed boundary layer and close enough to the aerosol sampling height (25 meters above sea level). Additionally, the chosen receptor height reduces the risk of surface contact in the trajectory calculations caused by rounding errors or interpolation.

# 2.4 Sea ice and open water

- Daily Arctic ice maps were downloaded from the U.S. National Snow and Ice Data Center database (https://nsidc.org/data, last accessed 2024-09-22) every day from 1991 through 2023. North of  $\approx 87^{\circ}$ N, a circular mask covers the irregularly shaped data gap around the North Pole caused by the ice-sensing satellites' orbit inclination and instrument swath. After 2007, improved satellite technology reduced this pole gap to >89°N. At each trajectory point, the ice map nearest in time was utilized to identify up to four pixels of this ice map within 50 km of the trajectory point. The average open-water information taken over these nearest pixels was added to the respective trajectory point.
- Within the periods of the nodes in the SOMs, probability frequency distributions (pdf) of open water under the back trajectories of the respective particle size distributions were accumulated. These pdfs yield estimates of aerosol-related ice conditions during the different large-scale flow conditions represented by the nodes. The information from the ice maps was also utilized directly in specific statistics of ice conditions  $\geq 85^{\circ}$  N by relating the number of pixels with given ice conditions, e.g.,  $\geq 20$  % open water, to the total number of pixels  $\geq 85^{\circ}$  N. By doing this for August and September of each of the 33 years 1991-2023, long-term statistics and trends of ice conditions relevant to the present study were constructed.

# 3 Results

- Six characteristic summer atmospheric Mean Sea Level Pressure (MSLP) patterns or circulation regimes were identified as results of the SOM analysis, which can be seen in Fig. 1a. The mean 10-m wind vectors associated with the circulation regimes can also be seen in Fig. 1b. The MSLP circulation patterns, corresponding anomalies, and wind vectors were calculated based on the August-September 1991-2018 average. The six circulation regimes represent distinct pressure conditions, particularly over Greenland, Alaska, and northern parts of Russia. The circulation regimes that occurred most commonly during 1991-2018 were nodes 2, 3, and 6 (see Fig. 2a), and those nodes were selected for further investigation.
- Circulation regime 2 is characterized by high pressure over Greenland and anomalously low MSLP over northern Eurasia (Fig. 1a and Appendix Fig. A2). In this regime, the central Arctic Ocean experienced a relatively strong airflow primarily from the direction of the Beaufort, Chukchi, and East Siberian Seas towards the Fram Strait and the Greenland Sea. The air mass was anomalously warm, as indicated by the roughly 2° C temperature anomaly at 850 hPa level, over the Beaufort, Chukchi, and East Siberian Seas (Fig. 3b, left side chart). However, this warm air mass did not extend to Greenland and Barents Seas, which have negative temperature anomalies at the 850 hPa level. Circulation regime 2 was also associated with an anomalously large amount of cloud water over the ice-covered Arctic Ocean (Fig 4a, left side chart), which was linked to enhanced net longwave radiation at the surface (Fig 4b, left side chart). The combination of large-scale temperature advection and radiative heating by the clouds was associated with anomalously high temperatures of 2m (T2m) over most of the Arctic Ocean (Fig. 4c, left side chart).

The main feature of circulation regime 3 was the anomalously low pressure over Greenland, which steers the airflow from the northern North Atlantic towards the North Pole and further towards the Canadian Arctic Archipelago (Fig. 1a-b and Appendix Fig. A2). This regime efficiently transported large amounts of heat and moisture from the northern North Atlantic to the Arctic (Fig. 3c, middle chart). The air mass at 850 hPa was anomalously warm over Greenland, Barents, and Kara Seas (Fig. 3b, middle chart). This also led to an anomalously high amount of cloud water and enhanced net longwave radiation over the Greenland and Barents Seas (Fig. 4a-b, middle chart), which, together with the warm air advection, explained the anomalously warm T2m conditions (Fig. 4c, middle chart).

Circulation regime 6 (Fig. 3a-c and Fig. 4a-c, right side charts) was characterized by weak pressure gradients and, thus, by very weak large-scale winds over the Arctic Ocean. Advection of heat and moisture is weak, meaning that the Arctic conditions were not affected mainly by large-scale horizontal transport, but the conditions were somewhat more locally driven. This regime was associated with an anomalously small amount of cloud water, which enabled enhanced cooling by the longwave radiation. In particular, the T2m was anomalously low (Fig. 4c, right side chart).

Figure 1: (a) Mean sea-level pressure and (b) wind vector at 10m height above ground level circulation regimes (2x3 nodes): calculated for August-September, including all years from 1991 to 2018.

**Figure 2: (a)** August to September circulation regimes of occurrence for the six nodes displayed in Figure 1; calculated for all years from 1991 to 2018, **(b)** the mean persistence of nodes (in days).

**Figure 3: (a)** Mean sea-level pressure in hPa, **(b)** the anomaly of temperature in  $^{\circ}$ C for 850 hPa, **(c)** vertically integrated moisture vectors in kg m<sup>-2</sup> s<sup>-1</sup> for circulation regimes (nodes) 2,3, and 6; were calculated for August-September, including all years from 1991 to 2018.

**Figure 4: (a)** total cloud water anomaly in kg m<sup>-2</sup>, **(b)** net LW radiation anomaly in W m<sup>-2</sup>, **(c)** and anomaly of temperature at the surface in ° C for circulation regimes (nodes) 2,3 and 6; were calculated for August-September, including all years from 1991 to 2018.

# 4 Linking particle size distributions to SOMs and conditions of "melt" and "freeze-up"

Whenever available, the hourly size distribution data of the five I/B Oden cruises were averaged  $\pm 6h$  around the SOM times before being grouped into the six SOM nodes. When the statistics of SOM occurrence were reduced to the five expedition years in Fig. 5, the dominance of nodes 2, 3, and 6 in Fig. 2 was maintained.

**Figure 5:** Probabilities of the occurrence of the six nodes in Figure 2 in August and September for the five expedition years: 1991, 1996, 2001, 2008, and 2018.

The I/B *Oden* expeditions all started in the developing "melt" and ended before "freeze-up" was complete (cf Appendix Fig. A1). Near-surface temperatures dropping below zero characterize the transition from "melt" to the start of "freeze-up." Tjernström et al. (2012) suggested using a threshold of - 2° C for this transition. The net surface energy balance could also indicate melting and freezing; as the surface temperature is practically at zero, a negative net surface energy balance indicates freezing, and a positive one provides energy for melting. However, in this study, the -2° C temperature threshold was augmented by adjusting it with onboard observations of ice formation during the individual cruises. With this procedure, hourly timelines were formed with conditions of "melt" and "freeze-up" for each expedition. The aerosol observations were then grouped according to the two ice conditions. The start of "freeze-up" was estimated to be August 18, 19, 19, 21, and 27 for 1991, 1996, 2001,2008, and 2018 data sets, respectively.

Statistics of the occurrence of the nodes in the two ice conditions were collected in Fig. 6. Figure 6a indicates that melt conditions were mainly associated with circulation regime 3. This was a consequence of the strong advection of heat and moisture from the North Atlantic and enhanced longwave radiative heating at the surface due to warm and moist air and excessive cloud water. The freeze-up was most commonly linked to circulation regime 6 as seen in Fig. 6b, where there was very weak horizontal transport of heat and moisture, and the meteorological conditions were more locally driven. Regime 6 enabled enhanced radiative cooling at the surface. The combination of low wind speeds, calm waters that limit the sea ice movement, and efficient radiative cooling created favorable circumstances for the freeze-up.

**Figure 6:** Probabilities of the occurrence of the six nodes in Figure 2 in August and September for the five expedition years: 1991, 1996, 2001, 2008, and 2018. The data is sorted into (a) "melt" and (b) "freeze-up".

In addition to melting and freezing (i.e., thermodynamic) conditions, the circulation regimes have dynamic impacts on sea ice. The pack ice in the central Arctic Ocean moves in response to air stress, water stress, sea surface tilt, and the Coriolis force. The geostrophic wind influences approximately half of the long-term ice motion, while the other half is attributed to mean ocean circulation. Over shorter time scales, more than 70 % of ice velocity variability can be accounted for by the geostrophic wind (Thomdike and Cheung, 1977).

The wind-driven Arctic ice circulation consists of the Beaufort Gyre, a clockwise circulation north of Alaska that spawns low winds, and the Transpolar Drift Stream (Timmermans and Marshall, 2020). The latter moves ice from Siberia across the Arctic basin to the North Atlantic off the east coast of Greenland. Sea ice transport towards the Fram Strait follows. Wind vector circulation regime 2 (Fig. 1b) can thus significantly affect the central Arctic Ocean's Sea ice concentration by inducing sea ice transport via the Fram Strait. On the other hand, the winds associated with regime 3 (Fig. 1b) tend to mechanically push and pack the sea ice towards the central Arctic Ocean and the coast of the Canadian Arctic Archipelago. However, the sea ice field responds relatively slowly to the atmospheric circulation field, and therefore, it was relevant to how persistent the circulation regimes were at the time. Persistent, long-lasting occurrences of circulation regimes have more potential to modify the sea ice field. As shown in Fig. 2b, regime 2 was the most persistent, lasting on average for 4 days, while regimes 3 and 6 typically prevailed for 3 days, calculated for all years from 1991 to 2018.

In Fig. 7, the pressure regimes of nodes 2, 3, and 6 were compared to average particle size distributions during "melt" and "freeze-up" of the respective nodes for data ≥ 85° N. Despite strongly differing regional meteorological conditions, all size distributions during "freeze-up" exhibited high particle concentrations in the sub-Aitken region below 30 nm and a secondary mode or at least a concentration shoulder above 100 nm diameter. The high sub-Aitken concentrations were missing during the melt in nodes 3 and 6. In contrast, the size distributions in node 2, also in "melt" indicated new particle formation with high nucleation mode concentrations below ca. 15 nm diameter, which will be discussed with below. The consideration is limited to new particle formation from the gas phase or from the division of sub-micrometer particles in their airborne state (Baccarini et al., 2020; Covert et al., 1996; Heintzenberg et al., 2006; Karl et al., 2013; Lawler et al., 2021; Leck and Bigg, 2010). A more detailed discussion will follow in section 5.4.

Figure 7: (a) Nodes 2, 3, and 6 surface pressure regimes, (b) Average particle size distributions during "melt" and "freeze-up" of the nodes in the left panel. Only aerosol data ≥85° N are displayed. Also displayed are error bars representing the standard deviations of the means of the average size distributions.

Figure 8: Sea ice coverage under five-day back trajectories in node 3, (a) "melt"; (b) "freeze-up". Only geocells with  $\geq$ 30 trajectory points are displayed. No ice data is close to the pole due to the ice-sensing satellites' orbit inclination and instrument swath.

The probability frequency distributions (pdfs) of open water conditions in Fig. 9 show the differences between nodes and ice conditions more clearly than maps. Figure 9a connects to the maps in Fig. 8. It showed very little solid ice in "freeze-up" compared to "melt" and high probabilities for broken-up ice with a distinct peak around 10 % open water. The corresponding pdf maximum lay at 5 % open water in "melt." Both ice conditions exhibit broad shoulders towards 50 % open water, albeit with higher probabilities in freeze-up. Completely open water under the back trajectories occurred in both conditions with about 20 % probability. The pdfs for nodes 2, 3, and 6 in Fig. 9b confirmed and emphasized the widespread occurrence of broken ice during freeze-up for node 3, as shown in Fig. 8b.

**Figure 9: (a)** Probability frequency distributions (pdfs) of open water under the back trajectories of node 3 in "melt" and "freeze-up", **(b)** As top panel but for "freeze-up" in nodes 2, 3, and 6, **(c)** As top panel but for node 2 and average conditions during the "melt" of nodes 1, 3, 4, 5, and 6. When the probabilities of 0% and 100% open water lie outside the scale of a graph, the respective values are given as numbers in text boxes.

The outlier size distributions of node 2 in Fig. 7b, which show a high concentration of nucleation mode remaining in the melt group, can now be understood by referring to the pdfs for "melt" in Fig. 9c, These pdfs represent the average of nodes 1, 3, 4, 5, and 6 and show an overall decrease from a 14% probability of solid sea ice to a low probability of broken ice beyond 50% open water, with a narrow probability extreme of 100% open water.

In node 2, the sea ice distribution was very different. Solid ice had a 32 % probability, and 100 % open water only 2 %. At the same time, broken ice occurred with frequencies of up to 10 % and a broad peak of around 30 % in open water. Inspecting the expedition years contributing to the aerosol data in the "melt" of node 2 revealed that almost exclusively, the year 1996 controlled the aerosol data in this node during "melt." The persistent, strong, high-pressure system over Greenland and the western Canadian Arctic Archipelago in 1996 led to the early formation of broken ice during "melt." This peculiar situation in 1996 was discussed by Nilsson and Barr (2001). In conclusion, the shape of particle size distributions, which exhibited high sub-Aitken concentrations, notably below 10 nm in diameter, indicated strong particle formation over the Arctic pack ice area ≥85° N. These particle sources appeared to be linked to the occurrence of broken ice during "freeze-up" - a condition most commonly associated with Node 6 (cf. Fig. 6b). The combination of low wind speeds, restricted sea ice movement, and effective radiative cooling contributed to favorable freeze-up conditions.

# 5 Atmospheric and ice conditions in the inner Arctic in summer and early autumn and long-term implications for regional aerosol sources

## 5.1 Trends of nodes and related ice conditions

SOMs were available for all years 1991-2018. With this more extensive dataset, the potential of the general occurrence of the most significant nodes was explored. Over the years analyzed, there was no systematic variation in the frequency of node occurrences (cf. Table A2 in the Appendix). Therefore, the following discussion disregarded their interannual variation.

As mentioned in the Introduction, Heintzenberg et al. (2015) showed that ice conditions with open water between 10 % and 30 % a few days before air mass arrival at the observation favored aerosol sources, of which the latter will be discussed further within section 5.4. Consequently, a lower limit of 20 % open water was adopted in the general discussion of aerosol sources over the inner Arctic. In the daily ice maps for August and September 1991-2018, the average number of pixels with open water of at least 20 % relative to the total number of pixels north of 85° N, termed "open water fraction" (OWF), was calculated and plotted as a black line in Fig. 10, which was extended to the end of 2023 for the subsequent discussion of possible trends. The interannual variability of OWF is substantial and appears to be stronger after the shift in 2006. Despite this high variability, the fraction of open water increases, at least when the time series is divided into two segments: before and after 2006 (cf. Table A2 in the Appendix). This division into two segments is also discussed in detail in Polyakov et al. (2023) describes an Arctic "switchgear mechanism" involving oceanic circulation. The five expeditions' average open water fractions during August and September, marked as filled yellow circles in Fig. 10, show that neither trend nor variability of the black curve could have been assessed with the expedition data only. The interannual variations of OWF are similar in all nodes (2, 3, and 6). Therefore, only the respective curve for the most prominent node, 3, is shown in Fig. 10 (orange line). Beyond that, segment averages of OWF of all three prominent nodes are marked as dotted and dashed lines, shown in Fig. 10. The segment ratios (before and after 2006) for nodes 2, 3, and 6 are 1.6, 2.1, and 1.6, respectively. The most significant ratio was observed for the wind-driven ice circulation associated with node 3 (cf. Fig. 1b), mechanically pushing and packing the sea ice towards the central Arctic Ocean and the coast of the Canadian Arctic Archipelago. As discussed in Chapter 3, this regime would, in addition, have effectively transported significant amounts of heat and moisture from the northern North Atlantic to the inner Arctic with potential impacts on sea ice melt and increased OWF (Mortin et al., 2016).

Figure 10: Average open water fractions  $\geq$  20 % Open Water (OWF in %)  $\geq$  85° N during August and September of the years 1991-2023 (full black line) and during regional Arctic circulation according to node 3 (full orange line). Average OWF values during August - September (8-9) and for nodes 2, 3, and 6 before and after 2006 are shown as dotted and dashed lines. Openwater conditions during the five I/B *Oden* cruises are indicated in full yellow circles.

Thus, the features of circulation regime 3 were suggested to explain not only the unusually warm (T2m) conditions shown in Fig. 4c (middle chart) but also the most significant change in the average open water fractions ( $\geq 20$  %) for latitudes  $\geq 85^{\circ}$  N during August and September before and after 2006. However, the average value of 1.7 for all three node segment ratios was close to the segment ratio of 1.8 for all August and September days. This similarity leads us to conclude that the different atmospheric circulation patterns of the most prominent nodes (2, 3, and 6) did not significantly lead to differing ice conditions in the inner Arctic, including all years from 1991 to 2018, which deviates from the findings of Thomdike and Cheung (1977) concerning the importance of the geostrophic wind for the movement of the sea ice. Therefore, the atmospheric circulation regimes cannot explain the substantial temporal changes and high interannual variability of ice conditions seen in Fig. 10. In Chapter 4, the locally determined freeze-up was decisive in initiating the conclusion of summer through early autumn, with strong new particle formation subsequently observed on I/B *Oden*.

Figure 11: Average values of open water fractions  $\geq 20$  % (OWT, full orange line), surface air temperatures (T2m, full black line), and sea surface temperatures (SST, full gray line), (a)  $\geq 85^{\circ}$ N and in marine regions (b) 78° N-82° N for all August/September months from 1991 to 2023. Averages  $\leq 2006$  and  $\geq 2007$  are drawn as dotted and dashed lines in the respective colors.

# 5.2 Trends in critical atmospheric and oceanic parameters

The analysis was extended over the whole inner Arctic (≥85° N) using the ice maps and average ERA5 temperatures, and it was extrapolated over all the years, from 1991 to 2023. In Fig. 11, average summer, including September T2m, SST, and OWF values, are collected in the two analyzed Arctic regions during 1991-2023. With substantial interannual variations, nearly all parameters increased with time in both geographic regions (cf. Table A2 in the Appendix). SST ≥85° N is the exception, for which ERA5 gives fixed values of -1.69° C, which is close to the freezing point of seawater of ca 35 ‰. Average levels in the two segments ≤2006, and ≥2007 are shown as dotted and dashed lines. In these segments, T2m increased on average by 0.7° C and 1.2° C in sectors ≥85° N and 78° N-82° N, respectively, whereas SST in the latter region increased by 0.2° C. The corresponding increases are 1.8 and 1.3, expressed as ratios in OWF, i.e., the open water fraction increased by 80 % and 30 %, respectively. Some correlations of the parameters in Fig. 11 are apparent. The highest values are reached in the region 78° N-

82 ° N with T2m and OWF showing an  $r^2 = 0.7$ . The following highest correlation concerns T2m and SST in the same region with an  $r^2 = 0.6$ .

## **5.3 Seasonal Changes in Ice Conditions**

 Employing the ERA5 T2m-data, "melt" and "freeze-up" were delineated according to the following schemes in the two studied geographic regions:

- Start of melt: Day of year (DOY) when regional average T2m rises over -1° C,
- End of melt (start of freeze-up): DOY when regional average T2m sinks below -2° C,
  - End of freeze-up: DOY when regional average T2m sinks below -10° C.

The threshold of -1° C for the start of "melt" follows the approach presented in Rigor (2000). As mentioned previously, the value of -2° C for the onset of "freeze-up" was suggested by Tjernström et al. and (2012). When the regional average surface air temperature is considered to be below the somewhat arbitrarily chosen -10° C, the completion of the freeze-up of leads is noted. The variation of the resulting DOY-values over the study period 1991-2023 allows the formulation of trends in the seasonality of the ice cover that may be relevant for regional aerosol sources.

For the region ≥85 °N, the temporal development of the three critical DOY-values is depicted in Fig. 12. Over the studied period of 33 years, climate warming yielded systematic trends with an earlier start of melting and later start and end of "freeze-up," albeit with substantial variabilities, being highest at the end of "freeze-up." As a result, the length of both "melt" and "freeze-up" increased with time. For the reference region 78° N-82° N, the directions of the trends are the same, albeit with different slopes. The trends are not statistically significant, but they are obvious.

Figure 12: Annual day-of-year-values (DOY) with onset of "melt" (average T2m >-1° C, full gray line), onset of "freeze-up" (average T2m < -2° C, full black line), and end of "freeze-up" (average T2m < -10° C, full orange line) for the region  $\geq$ 85° N. Respective linear trends are shown as dotted and dashed lines.

Table 2 collects the changes in critical DOY values and length of "melt" and "freeze-up" from 1991 to 2023 for assumed linear developments in the two studied Arctic regions. In 2023, "melt" started  $\approx 5$  days earlier than 1991 for latitudes  $\geq 85^{\circ}$  N; "freeze-up"  $\approx 5$  days later, ending 16 days later, yielding an increase of  $\approx 9$  days for melt and  $\approx 12$  days for "freeze-up". Further south, the shifts in critical DOY-vales are more substantial in the reference region. Consequently, the length of "melt" nearly doubled, whereas the length of "freeze-up" is somewhat shorter than further north.

**Table 2:** Changes in critical DOY-values and lengths of "melt" and "freeze-up" from 1991-2023 assuming linear developments in the two regions  $\geq$ 85° N, and 78° N-82° N.

| Region      | Shift of "melt" onset (days) | Shift of onset<br>of "freeze-<br>up" (days) | Shift of end<br>of "freeze-<br>up" (days) | Extension of "melt" (days) | Extension of<br>"freeze-up"<br>(days) |
|-------------|------------------------------|---------------------------------------------|-------------------------------------------|----------------------------|---------------------------------------|
| ≥85° N      | -4.6                         | 4.7                                         | 16.3                                      | 9.3                        | 11.6                                  |
| 78° N-82° N | -5.3                         | 12.1                                        | 22.4                                      | 17.4                       | 10.2                                  |

## 5.4 Long-term implications for central Arctic aerosol sources

Before exploring the future implications of the observed trends for the central Arctic aerosol, this section summarizes the seasonal variation in particle size distributions, including all available data beyond the SOMS discussion. Figure 13 presents the averages and medians of all data collected at  $\geq$  85° N during the five I/B *Oden* expeditions, covering both "melt" and "freeze-up' periods. The figure highlights significant differences associated with the two ice conditions shown in Fig. 7.

The "freeze-up" samples, attributed to Figure 13, typically stayed in the air over the pack ice area for over five days before being collected at latitudes of 85° N or higher (e.g., Leck and Svensson, 2015). Extended advection over pack ice has been observed to result in comparatively low particle concentrations for diameters larger than approximately 80 nm, due to scavenging in low clouds and fog, especially during the first 1-2 days of advection from the open sea into the pack ice. (Heintzenberg et al., 2006; Nilsson and Leck, 2002). The tiniest particles, with diameters below 30 nm, have been observed to have very short atmospheric lifetimes, generally ranging from hours to a day (Leck and Bigg, 1999). Therefore, their presence over the inner pack ice cannot be explained by advection from more southerly sources.

The somewhat similar median distributions during "melt" and "freeze-up" were interpreted as representing the inner Arctic background, whereas individual particle formation events strongly influenced the averages. Particles with diameters under 30 nm, as shown in Figure 13, had particle number concentrations during "freeze-up" that were more than two orders of magnitude higher than during "melt," especially for particles under 10 nm, indicating strong new particle formation. Based on aerosol particle number size distributions measured on I/B *Oden* covering the months of August and September of 1991, 1996, 2001, 2008, and 2018, a common characteristic of individual particle formation events is that the particle concentrations under 10 nm in diameter are often very low. Still, they can suddenly rise dramatically for 5-12 hours, reaching concentrations of several hundred to 1000 cm<sup>-3</sup> in a background atmosphere with very low total aerosol numbers, typically around 100 cm<sup>-3</sup> or less than 10 cm<sup>-3</sup> (Covert et al., 1996), with weak subsequent growth before being scavenged by fog or rain (Karl et al., 2013; Leck

and Bigg, 1999; Baccarini et al., 2020). Events with elevated 3–5 nm particles also show increased concentrations in other size ranges, less than about 30-50 nm, reaching up to 500 cm<sup>-3</sup> for several hours (Leck and Bigg, 1999;2010; Karl et al., 2013). The occurrence of the events is especially notable during the freeze-up period.

The formation of numerous small particles below 10 nm in diameter is likely due to homogeneous nucleation originating from gaseous precursors, including iodic and sulfuric acids. These acids yield initial particle clusters that grow further by condensation, potentially supported by iodine acid or biogenic organic compounds vapors, or as a combination of production via the generation of marine polymer gels, which are released as small nanometer-sized (nano-granular) particles when clouds or fog droplets dissipate (Baccarini et al., 2020; Heintzenberg et al., 2006; Karl et al., 2013; Lawler et al., 2021; Leck and Bigg, 1999; 2010). The average number concentration of a prominent broad peak during "melt" was reported to involve emissions of biogenic particles, especially polymer gels, from the MIZ or open leads over the pack ice, and growth of pre-existing smaller particles through heterogeneous condensation of precursor gases like sulfuric and methane sulfuric acids from photochemical oxidation of DMS and aerosol cloud processing (e.g. Leck and Bigg, 2005b). As noted above, the very low aerosol concentrations over 300 nm diameter were shown to result from efficient scavenging near the MIZ.

**Figure 13:** Average and median particle size distributions at ≥85°N collected over the years 1991, 1996, 2001, 2008, and 2018 during the "melt" and "freeze-up" phases. Also displayed are error bars representing the standard deviations of the means of the average size distributions and median absolute deviations of the median size distributions.

Over the study period of 33 years, several atmospheric and oceanic parameters relevant to regional aerosol sources showed significant changes consistent with the known Arctic warming. Assuming an overall linear change (instead of the segment changes  $\leq 2006/ \geq 2007$ ), T2m increased by 1.1° C  $\geq 85^{\circ}$  N and 2° C in the marginal ice region 78° N-82° N.

Moreover, during the 33 years, this study's results show that OWF nearly doubled  $\geq$ 85° N while it increased by  $\approx$ 50 % in the region 78° N-82° N. In the latter region, SST increased by 0.4°C, assuming a linear trend; "melt" increased in length by more than a week  $\geq$  85° N, increasing open water areas in sea ice (leads), and by more than two weeks in the region 78° N-82° N.

According to Aslam et al. (2016), these changes must influence sea ice distribution, such as open water and newly formed leads. As a result, the response of microorganisms in seawater to melting or freezing ice could have an impact on various biogenic sources at the air-sea interface.

The overall thinning of sea ice, along with earlier and longer melt periods, clearly results in more open water. This increases the sunlight reaching the ocean surface and promotes phytoplankton growth, utilizing the nutrients already available and those supplied by melting ice-bottom algae (Arrigo et al., 2012). This would also regulate polymer gel production via phytoplankton secretions, as reviewed by Deming and Young (2017) and references therein. However, diminished ice thickness or increased openness of the sea would facilitate more efficient wind mixing of the surface ocean, thereby augmenting the depth of the mixed layer and potentially mitigating algal growth. Because of polymer gels that induce aggregation (Orellana et al., 2011), increased carbon flux from sea ice might occur with earlier ice melt at the MIZ if grazers feeding on aggregates are absent, resulting in less accumulation of polymer gels in the upper water column (Carmack and Wassmann, 2006).

The extension of "freeze-up" by about ten days with freshly frozen leads restricts the exchange of nascent sea spray particles with the atmosphere. However, Bowman and Deming (2010) discovered that frost flowers contain significantly more bacteria and extracellular polymer gels than brines, young ice, or water. Their research indicates that an increase in frost flower occurrence could promote chemical interactions between sea ice and the atmosphere, serving as an enriched atmospheric source of polymer gels as well as for iodine released via the frost flowers from sea-ice brine channels.

Whereas the sum effect of counteracting processes during "melt" on the biogenic Arctic aerosol in a warming climate is unclear, the net impact of the changing "freeze-up" is expected to enhance the biogenic Arctic aerosol in late summer/autumn. In terms of particle size distribution, this may lead to an even more prominent sub-Aitken mode than shown in Fig. 13.

Results from large eddy simulation models indicate that Aitken mode particles could significantly influence the cloud's simulated microphysical and radiative properties by forming cloud droplets (Bulatovic et al., 2021). Their findings aligned with aerosol particle size distribution data collected during the five-year I/B *Oden* expeditions, which showed that Aitken particles as small as approximately 50-30 nm in diameter can act as CCN. These particles also showed an increased tendency to activate into cloud droplets after the commencement of sea ice formation (Duplesee et al., 2024; Karlsson et al., 2022; Leck and Svenson, 2015). Based on the findings outlined above, the response of microorganisms in seawater to the processes of melting or freezing of ice could significantly impact the formation of low-altitude liquid clouds within the high Arctic environment through aerosol-cloud interactions. This, in turn, may have implications for their radiative properties and the future evolution of the ice cover.

## 6 Conclusions

The starting point of the present study was aerosol particle number size distributions measured ≥85° N on five cruises of I/B *Oden* covering the summers of 1991, 1996, 2001, 2008, and 2018, and previous analyses indicating different potential source regions and ice-related factors affecting Arctic aerosol sources. Regional atmospheric circulation regimes (nodes) based on the method of self-organizing maps (SOMs) were investigated as potential controllers of Arctic aerosol sources. Circulation regime 2 featured high pressure over Greenland and low MSLP in northern Eurasia. The central Arctic Ocean experienced strong airflow from the Beaufort, Chukchi, and East

Siberian Seas towards the Fram Strait and Greenland Sea. Circulation regime 3 displayed low pressure over Greenland, directing the airflow from the northern North Atlantic to the North Pole and Canadian Arctic. Circulation regime 6 showed weak pressure gradients, causing extremely light large-scale winds across the Arctic Ocean. Despite substantial climate change, the three most prominent nodes were not connected to regional source-related differences and did not vary systematically throughout the study period. Instead, the seasonal course of sea ice melt and freeze-up appeared to affect the shape of the aerosol size distributions significantly. In particular, high sub-Aitken concentrations occurred during the "freeze-up", most commonly associated with the low wind, restricted sea ice movement, and effective radiative cooling conditions of node 6. The high concentrations of newly formed particles measured during "freeze-up" were interpreted as deriving from frost flower formation during this time of the year.

Based on the understanding that ice conditions and their seasonal course are considered major controllers of Arctic aerosol sources, the study was extended to cover all years from 1991 to 2023 to enable speculations about changing aerosol source conditions in the warming Arctic climate. With daily ice maps and sea surface and atmospheric temperatures from the ERA5 database, long-term changes in ice conditions were explored. Over the 33 years of the study, the significant increases in sea and air temperatures nearly doubled the favorable ice conditions for new particle formation ≥85° N, lengthening both "melt" and "freeze-up" parts of the illuminated Arctic by more than a week. Whereas the sum effect of counteracting processes during the ice melt season on the airborne biogenic Arctic aerosol in a warming climate is unclear, the net effect of changing the freeze-up of sea ice is expected to enhance the airborne biogenic Arctic aerosol in late summer/autumn. The consequences of the foreseen seasonal changes in biogenic aerosol sources in the inner Arctic remain to be investigated. The strong aerosol-cloud-climate correlation necessitates regional model simulations to evaluate potential future impacts of a doubling in airborne biogenic particles during the freeze-up period and an indeterminate net source change in the melt season.

# Appendix A

**Table A1:** Percentage of unfitted data, average absolute and relative deviations of the lognormally fitted distributions from the measurements for the five expedition years.

| Year | Unfitted<br>data (%) | Av. abs.<br>Dev. | Av. rel.<br>Dev. |
|------|----------------------|------------------|------------------|
| 1991 | 9                    | 0.27             | 0.23             |
| 1996 | 3.7                  | 0.21             | 0.24             |
| 2001 | 0.7                  | 0.22             | 0.23             |
| 2008 | 4                    | 0.28             | 0.23             |
| 2018 | 8                    | 0.19             | 0.24             |

**Table A2:** Statistics and two-tailed statistical tests of significant changes ≤2006 versus ≥2007 of node occurrence (node 1-6), average August/September temperature (T2m) ≥85° N, and 78° N-82° N, sea surface temperature (SST) 78N-82°N, open water fraction (OWF) ≥ 20 % ≥ 85° N, and 78° N-82° N, median value of first day-of-year (DOY) with average T2m ≥ 85° N >-1 °C, median value of first day-of-year (DOY) with average T2m 285° N sinking < -2° C, median value of latest day-of-year (DOY) with average T2m 78° N-82° N sinking < -10° C, median value of latest day-of-year (DOY) with average T2m 78° N-82° N sinking < -10° C, length of melt period ≥ 85° N, length of melt period 78° N-82° N, length of freeze-up period ≥ 85° N, length of freeze-up period 78° N-82° N. The changes are considered significant if  $P(\le t)$  two tail is less than 5 %.

| Parameter               | Unit | Mean<br>≤2006 | Variance<br>≤2006 | Mean<br>≥2007 | Variance<br>≥2007 | P(T≤t)<br>two tail | Change significant? |
|-------------------------|------|---------------|-------------------|---------------|-------------------|--------------------|---------------------|
| Node 1                  | n.a. | 0.127         | 0.008             | 0.150         | 0.007             | 0.560              | no                  |
| Node 2                  | n.a. | 0.251         | 0.013             | 0.253         | 0.015             | 0.970              | no                  |
| Node 3                  | n.a. | 0.228         | 0.005             | 0.194         | 0.015             | 0.400              | no                  |
| Node 4                  | n.a. | 0.099         | 0.002             | 0.067         | 0.003             | 0.096              | no                  |
| Node 5                  | n.a. | 0.100         | 0.003             | 0.068         | 0.003             | 0.110              | no                  |
| Node 6                  | n.a. | 0.193         | 0.011             | 0.199         | 0.010             | 0.870              | no                  |
| T2m_8-9, ≥ 85° N        | C    | -4.5          | 16.7              | -3.6          | 12.5              | 3.32E-13           | yes                 |
| T2m_8-9, 78° N-82° N    | C    | -3.3          | 9.0               | -2.1          | 4.6               | 3.03E-44           | yes                 |
| SST, 78° N-82° N        | C    | -1.5          | 0.005             | -1.3          | 0.012             | 9.70E-07           | yes                 |
| OWF, ≥ 20 % ≥ 85° N     | %    | 22            | 138               | 40            | 267               | 1.60E-03           | yes                 |
| OWF, ≥ 20 % 78° N-82° N | %    | 62            | 54                | 81            | 42                | 5.50E-9            | yes                 |
| DOYmin, > -1° C ≥ 85° N | DOY  | 162           | 36                | 160           | 41                | 0.370              | no                  |
| DOYmin, 78° N-82° N     | DOY  | 163           | 16                | 161           | 29                | 0.170              | no                  |
| DOYmax,

**Figure A2:** Anomaly of mean sea-level pressure in hPa for circulation regimes (nodes) 2, 3, and 6, calculated for August-September of all years from 1991 to 2018.

- Data availability: The datasets used and/or analyzed during the current study are available on reasonable request
- from the authors Caroline Leck (lina@misu.su.se), Jost Heintzenberg (jost@tropos.de), and Tuomas Naakka
- (tuomas.naakka@fmi.fi). The subset of observations from the 2018 expedition used in this study can be accessed
- through the Bolin Centre Database (<a href="https://doi.org/10.17043/">https://doi.org/10.17043/</a> oden-ao-2018-aerosol-dmps-1, Karlsson and Zieger,
- 2020; https://doi.org/10.17043/oden-ao-2018-misu-weather-3.
- **Competing interests:** The authors declare that they have no conflict of interest.
- Disclaimer: Special issue statement: the statement on a corresponding special issue will be included by
- Copernicus, if applicable.
- Author contribution: CL Conceptualization, Methodology, Validation, Formal analysis, Investigation,
- Resources, Writing Original Draft, Writing Review & Editing, Visualization, Project administration. JH -
- Methodology, Validation, Formal analysis, Data Curation, Writing Original Draft, Writing Review & Editing,
- Visualization. TN Validation, Formal analysis, Data Curation, Writing Parts of Original Draft Writing -
- Review & Editing, Visualization. TN Data Curation, Writing Parts of Original Draft, Writing Review &
- Editing.

- Acknowledgments: We thank John Ogren for his generous assistance in acquiring the 2023 ice data following a
- significant alteration in the NSIDC file structure. We thank Pasi P. Aalto, Douglas Orsini, Paul Zieger, and Andreas
- Baccarini for their dedicated efforts in physical aerosol data collection and quality assurance. We acknowledge
- the Swedish Polar Research Secretariat (SPRS) for facilitating access to the Swedish icebreaker *Oden* and
- providing logistical support. Finally, we wish to convey our deep appreciation to the Captains of the *Oden* during
- the five research expeditions in 1991, 1996, 2001, 2008, and 2018: Anders Backman, Mats Johansson, Mattias
- Peterson—and their exemplary crew for their invaluable assistance.
- Financial support: This work was supported by the International Meteorological Institute (IMI), the Swedish
- Research Council (VR), the Knut and Alice Wallenberg Foundation, and the Bolin Centre for Climate Research
- at Stockholm University.
- Review statement: the review statement will be included by Copernicus.

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
