# Peer review of "Is the summer aerosol over the Arctic controlled by regional 1"

_EGUsphere, 2025_

## Referee Comment (RC2)

Review: "Is the summer aerosol over the Arctic controlled by regional atmospheric circulation or ice conditions? Trends and Future Implications"
Leck et al.

This paper combines in situ shipboard measurements of high Arctic aerosol size distributions spanning the years 1991-2018 with reanalysis and back-trajectories to relate summer synoptic process and sea ice conditions on changes to the size distributions. The authors find, using Self Organizing Maps, that three synoptic regimes were most common for August-September and likely have the greatest effect on the measured distributions. Namely, the largest differences were between freeze-up and melting conditions, such that new particle formation from frost flowers were likely attributable to size distribution changes. Using the observed regime shift in open water fraction and temperature during the record, the authors posit that future conditions will become more favorable for new particle formation in the high Arctic, however, diverse impacts from different biogenic aerosol sources at the air-sea interface due to these changes confounds predicting exact changes.

Overall, this paper provides interesting insights on environmental and meteorological conditions that likely impact size distributions and the implications for these effects in the future, however several areas of the discussion lack clarity, depth, and support from the available measurements and the analysis conducted. The authors should make a concerted effort to carefully revise and edit the paper for clarity and continuity between passages and sections. I believe this paper is suitable for this journal, but it is necessary to consider and address the following, major comments and technical concerns before being acceptable for publication.

Major Comments:
- There are several passages in this paper that speak at depth to the biological and physical characteristics at the ocean surface and its composition, but many of these are ancillary to the key points made in the paper and their relation to the overall findings. In the proceeding major/technical comments I identify where some of these sections are and how they lead to a digression in the focus and should thus be reduced or revised to more directly emphasize their relevance.

- Introduction: I believe this section should be significantly revised. The first couple of paragraphs in the introduction can be reduced for brevity. Too much superfluous detail is included here that does not exactly connect to the main motivation or research questions that are meant to be answered in the paper. It would be much more appropriate for the authors to include discussion on relevant Arctic topics including air mass characterizations, the seasonality of aerosol and their sources, and potential impacts of synoptic circulation and regional processes. Less attention should be focused on jet/film/gels etc. It is necessary that the authors provide support and context that justifies the paper's motivation. What are the current limitations and uncertainties and research questions that motivate carrying out this work?

- SOM Clustering: How can the authors justify that the use of only August-September in the SOM sufficiently captures synoptic variability in the Arctic? I believe limiting the

scope to these months will characterize the synoptic circulation most relevant and common to those months, but likely not the extensive annual and interannual variability throughout the season that could occur. Was the full annual cycle tested for comparison? I feel that completing such an analysis will justify this choice and make the results more robust.

- More detail should be provided on the lognormal fitting of the aerosol size distributions. Was this an automated process? How were the number of modes determined and the robustness of fit evaluated? The authors should provide statistics on the mode fits. What was done for periods in which an extension of the size range with the spectrometer wasn't included and how can those periods be justifiably compared to when only the DMA was available? All of these points must be discussed because the results of fitting of the lognormal modes is not discussed anywhere in the paper.

Minor Comments and Technical Corrections:
- Line 88: The more canonical term is "aerosol-cloud interactions", rather than "aerosol-to-cloud interactions". I suggest this be revised.

- Line 88-89: The authors should clarify how aerosol-cloud interactions counteract warming in the Arctic as relevant to this claim and citation.

- Line 95: Revise "most significant" to "largest".

- Lines 130-131: It should be specified here that the August-September input data were used because they coincided with when shipboard measurements were most often collected.

- Lines 140-141: Can the authors please be more specific when they say, "…sufficiently represent the range of large-scale circulation patterns in the Arctic."? There is no context for Arctic circulation patterns (in the introduction or elsewhere) prior to this description and no citation.

- Line 144-149: The citations that describe the ERA5 reanalysis and the description of its data and assimilation should be moved to the first mention of the product in Lines 125-126. In its current location it is a digression.

- Lines 150-152: Did the authors intend to say that means (averages) of the variables listed in these lines "were separately calculated" for each MSLP regime? If that is the case, I believe that this sentence can be rewritten for clarity to say that means of variables including […] were calculated for each MSLP regime.

- Line 156: Please clarify why the latitude band 78N-82N was used separately to 85N. What distinguishes these regions?

- Lines 163-165: The authors should specify what kind of differential mobility analyzers (and CPCs) were used to measure the aerosol size distributions. What were the size ranges of the DMAs?

- Line 166: What type of "contamination"? Contamination from the ship? Please specify and provide more detail (at least as a brief summary) of how and why this was done.

- Line 168: The authors should specify what kind of aerosol spectrometer was used and what was its size range.

- Line 169-171: For the "harmonized" size distribution interpolation, was any merging done? If so, please describe how and if there was diameter overlap between the different measurements. Please include detail on how considerations were made to combine distributions from mobility and/or optical diameters. An assessment of the merging should also be detailed.

- Line 189: What is meant by the sentence "Vertical layers one through 25 hPa separates five layers"?

- Figure 6: The size distributions in these plots show values extending down to 1 nm, while it was described in the methods that the distributions were interpolated to a range of 3.37-900 nm. Can the authors please clarify this discrepancy?

- Line 301-302: A citation or evidence should be provided here to justify claiming the source of the high nucleation mode as new particle formation in this region. The authors should also clarify what is meant by "new particle formation". I believe the authors mean the formation of particles from gaseous precursors. If so, this should be specified.

- Line 302: revise "dealt with" to "discussed".

- Figure 6: What is meant by the y-label in this figure? Are these size distributions showing the number concentration in each bin not the normalized number concentration per dlogDp bin? If it is the former, why is that presentation of the distribution shown rather than the dN/dlogDp (the concentration normalized by the differential log diameter which is more commonly used?

- Section 4: Can the authors provide some quantitative statistical information on the modal properties of the size distributions between nodes and melting/freeze-up? Specifically, the authors should quantify the mean diameters and number concentrations of the modes between nodes and freeze-up and melt. How do these properties compare to previous literature if available for synoptic regime differences in the size distribution? What do these results suggest for changes due to aerosol sources, synoptic circulation, and sea ice conditions?

- Line 323: Clarify what is meant by "outlier size distributions of node 2 in Fig. 6b." I believe the authors are indicating the high concentration nucleation mode remaining in the melt group.

- Lines 325-326: Specify that the description "Figure 8c showed an average…" is pertaining to the average of nodes 1,3,4,5, and 6.

- Line 326: Revise to "the [sea] ice".

- Lines 372-373: Please clarify what are the "segment ratios".

- Line 379-380: The final sentence in this passage is very confusing and vague. Please clarify/revise or remove.

- Lines 410-411: "systematic trends", what is meant by this? Were the trends found to be statistically significant?

- Lines 435-437: In these lines, the differences in the size distributions should be further quantified and compared using statistical information from the modes.

- Lines 437-439: Can the authors not attribute any of the differences in the size distributions between periods to transport? No evidence in this paper is given to justify claiming these differences are solely driven by new particle formation. A further consideration of these differences needs should be done here.

- Line 451-470: I believe this entire passage should be significantly reduced to remain within context and on topic of the discussion in this section. As it stands, this passage is a substantial digression. Key findings from the works and discussion provided here would be more useful to put into context for the proceeding paragraphs.

- Lines 471-502: Like my previous comment, there appears to be more superfluous content here that digresses from the main point of this section. The authors should make a more concerted effort to relate brief key findings from the many cited works in these passages within the context of their effect on aerosol size distribution changes in the long-term in freeze-up and melt conditions. Only a brief discussion from Lines 498-502 does this, but it is inadequate and should thus be expanded. One point of expansion could be discussing the size-dependent effect of these changes. For example, the authors discuss effects on sea spray sources and emission and cloud processing, which will likely have dissimilar effects to biogenic impact on sub-Aitken-mode particles and other modes.

- Lines 527-528: I don't think "aerosol-cloud-climate connections would [*require*] model simulations". Aircraft, ground-based, and remotely sensed cloud microphysical properties can be retrieved for this purpose by relating them to the observed aerosol measurements of this study and has been done in previous work that should be cited. This should also be mentioned in this concluding discussion.

---

## Author Comment (AC1)

**Referee 2**

Review: "Is the summer aerosol over the Arctic controlled by regional atmospheric circulation or ice conditions? Trends and Future Implications"

Leck et al.

This paper combines in situ shipboard measurements of high Arctic aerosol size distributions spanning the years 1991-2018 with reanalysis and back-trajectories to relate summer synoptic process and sea ice conditions on changes to the size distributions. The authors find, using Self Organizing Maps, that three synoptic regimes were most common for August-September and likely have the greatest effect on the measured distributions. Namely, the largest differences were between freeze-up and melting conditions, such that new particle formation from frost flowers were likely attributable to size distribution changes. Using the observed regime shift in open water fraction and temperature during the record, the authors posit that future conditions will become more favorable for new particle formation in the high Arctic, however, diverse impacts from different biogenic aerosol sources at the air-sea interface due to these changes confounds predicting exact changes.

Overall, this paper provides interesting insights on environmental and meteorological conditions that likely impact size distributions and the implications for these effects in the future, however several areas of the discussion lack clarity, depth, and support from the available measurements and the analysis conducted. The authors should make a concerted effort to carefully revise and edit the paper for clarity and continuity between passages and sections. I believe this paper is suitable for this journal, but it is necessary to consider and address the following, major comments and technical concerns before being acceptable for publication.

**We thank the reviewer for his/her very thoughtful, constructive, and helpful comments. We have incorporated our responses into the revised manuscript, which will be detailed below.**

**Major Comments:**

- There are several passages in this paper that speak at depth to the biological and physical characteristics at the ocean surface and its composition, but many of these are ancillary to the key points made in the paper and their relation to the overall findings. In the proceeding major/technical comments I identify where some of these sections are and how they lead to a digression in the focus and should thus be reduced or revised to more directly emphasize their relevance.

- I believe this section should be significantly revised. The first couple of paragraphs in the introduction can be reduced for brevity. Too much superfluous detail is included here that does not exactly connect to the main motivation or research questions that are meant to be answered in the paper.
It would be much more appropriate for the authors to include discussion on relevant Arctic topics including air mass characterizations, the seasonality of aerosol and their sources, and potential impacts of synoptic circulation and regional processes. Less attention should be focused on jet/film/gels etc. It is necessary that the authors provide support and context that justifies the paper's motivation. What are the current limitations and uncertainties and research questions that motivate carrying out this work?

  The Introduction section has been rewritten (see page 16) esssentially meeting the recommendations from Ref. 2.

  In the present study, the question of whether regional-scale atmospheric circulation patterns or the temporal development of the "melt"/"freeze-up" season plays a key role in summer aerosol sources over the inner Arctic is pursued. As a basis, we are using a synopsis of all aerosol number size distributions accumulated during five I/B *Oden* expeditions, 1991-2018, to the inner Arctic during August and September.

  We emphasize that all expeditions were summer studies, not directed or capable of addressing seasonal variations. Therefore, we do not see how a discussion about the annual seasonality of air mass characteristics, as well as aerosol number and mass and their sources, would help in achieving the primary objectives of the study.

  Nevertheless, we prefer to adhere to the established knowledge gained from the five expeditions conducted, combined with a discussion pertinent to the months of August and September concerning air mass characterizations, the seasonality of aerosols and their sources, and the potential impacts of synoptic circulation and regional processes. We agree with the reviewer that a full seasonal review would improve clarity but there simply is not the necessary database available yet.

- How can the authors justify that the use of only August-September in the SOM sufficiently captures synoptic variability in the Arctic? I believe limiting the scope to these months will characterize the synoptic circulation most relevant and common to those months, but likely not the extensive annual and interannual variability throughout the season that could occur. Was the full annual cycle

tested for comparison? I feel that completing such an analysis will justify this choice and make the results more robust.

We conducted an additional SOM analysis to examine the effect of the number of nodes and the choice of period (whole year vs. two months, August - September) on SOM clustering (included at the end of this document). The analysis revealed (see pages 12-15) that the occurrence of nodes clearly distinguishes circulation patterns between winter and summer. Since our study focuses solely on summer conditions, using SOM clustering with mean sea level pressure data from the entire year is not advantageous for the concise presentation of circulation patterns, because patterns common in winter are rare in summer and vice versa. Additionally, focusing on circulation patterns during the study period, August and September, allows us to use fewer nodes while still capturing detailed circulation features. Six nodes for August–September appear sufficient to capture the main characteristics of large-scale circulation. The differences between neighbouring nodes in the 12-node SOM analysis for August–September are relatively small, and using six nodes already provides an accurate detail of the circulation pattern that is obtained with only six nodes.

- More detail should be provided on the lognormal fitting of the aerosol size distributions. Was this an automated process? How were the number of modes determined and the robustness of fit evaluated? The authors should provide statistics on the mode fits. What was done for periods in which an extension of the size range with the spectrometer wasn't included and how can those periods be justifiably compared to when only the DMA was available? All of these points must be discussed because the results of fitting of the lognormal modes is not discussed anywhere in the paper.

We apologize for the lack of detail and extended the text to: The lognormal fitting was accomplished with an algorithm written in FORTRAN that follows the procedure described by Whitby and McMurry (1997).  The multimodal character of the sub-micrometer aerosol size distribution is well established in the summer Arctic. Thus, with two to six lognormal modes a given number size distribution is approximated over the total diameter range 1-1000 nm by randomly varying positions, logarithmic standard deviations, and total number concentrations of the modes.  When an optimal fit is arrived at, the number of modes is reduced as much as possible while keeping the average difference between model and measurement below a given upper limit.  Absolute concentrations should be maintained within 50% of the measurements while relative differences between model and measurements are maintained within 25%.  The latter condition secures a good simulation of the wings of the particle size distribution.  Statistics

of the five expeditions in Table S1 show that on average the quality of the lognormal approximations are substantially better than given by the constraints of the algorithm and less than 10% of the could not be approximated within the set limits.

**Table A1:** Percentage of unfitted data, average absolute and relative deviations of the lognormally fitted distributions from the measurements for the five expedition years.

| Year | Unfitted data (%) | Av. abs. Dev. | Av. rel. Dev. |
|------|------|------|------|
| 1991 | 9 | 0.27 | 0.23 |
| 1996 | 3.7 | 0.21 | 0.24 |
| 2001 | 0.7 | 0.22 | 0.23 |
| 2008 | 4 | 0.28 | 0.23 |
| 2018 | 8 | 0.19 | 0.24 |

**Minor Comments and Technical Corrections:**

- Line 88: The more canonical term is "aerosol-cloud interactions", rather than "aerosol-to-cloud interactions". I suggest this be revised.
  We have changed *"aerosol-to-cloud interaction" to "aerosol-cloud interactions".*

- Line 88-89: The authors should clarify how aerosol-cloud interactions counteract warming in the Arctic as relevant to this claim and citation. "It may help counteract some of the warming effects over the Arctic pack ice area (Mauritsen et al., 2011)."
  We have added the following for clarification in the rewritten Introduction section: Low-altitude liquid clouds are particularly important in Arctic climate change. By influencing the surface energy budget, these clouds can partially offset regional warming. These clouds in the summer high Arctic north of 80° contain fewer but larger droplets than clouds in other regions (Mauritsen et al., 2011). Coupled with the semi-permanent ice cover, even small changes in either can significantly influence heat transfer to the ice and its melting process. As such, the regional aerosol and its sources over the pack ice potentially play a significant role in regulating the surface energy budget through aerosol-cloud interactions. Ceteris paribus, if more aerosols become available for water uptake, clouds may form with numerous smaller droplets. This increases their sunlight reflection, leading to surface cooling (Twomey, 1974).

- Line 95: Revise "most significant" to "largest".
  *We have changed "most significant" to "largest".*

- Lines 130-131: It should be specified here that the August-September input data were used because they coincided with when shipboard measurements were most often collected.
  *The text now states: The MSLP data were collected at 12-hour intervals, covering days from August to September 1991–2018, and served as input for the SOM analyses. These input data were chosen because they aligned with the periods when shipboard measurements were most frequently available.*

- Lines 140-141: Can the authors please be more specific when they say, "...sufficiently represent the range of large-scale circulation patterns in the Arctic."? There is no context for Arctic circulation patterns (in the introduction or elsewhere) prior to this description and no citation.
  *The following clarification was made to the lines 140 – 141:  "These six nodes in the 2 x 3 array sufficiently represent the range of large-scale circulation patterns in the Arctic." Was replaced by: " These six nodes in the 2 x 3 array can adequately represent the range of large circulation patterns in the Arctic, so their variation of large circulation patterns in the Arctic is captured in the variation of the circulation patterns of the nodes in enough detail for the aims of the study.*

- Line 144-149: The citations that describe the ERA5 reanalysis and the description of its data and assimilation should be moved to the first mention of the product in Lines 125-126. In its current location it is a digression.
  *The description of the ERA5 analyses has been moved to Lines 125-126. The text now reads: The European Centre for Medium-Range Weather Forecasts (ECMWF) Reanalysis v5 (ERA5; Hersbach et al., 2020;Hersbach, 2023, (last accessed 2024-10-31)) mean sea level pressure (MSLP) fields were clustered using the Self-Organizing Maps (SOM) method to identify the Arctic's main large-scale atmospheric circulation regimes. ERA5 is a state-of-the-art global atmospheric reanalysis that applies a four-dimensional variational data assimilation method to assimilate various observations.*

- Lines 150-152: Did the authors intend to say that means (averages) of the variables listed in these lines "were separately calculated" for each MSLP regime? If that is the case, I believe that this sentence can be rewritten for clarity to say that means of variables including [...] were calculated for each MSLP regime.

The sentence is now rewritten for clarity and reads:
In this study, the means of variables, including 10-meter wind vectors, temperatures (at 2 meters and 850 hPa), vertically integrated moisture vectors, total cloud water, net longwave (LW) radiation, and surface temperature in °C were calculated separately for each of the six MSLP circulation regimes.

- Line 156: Please clarify why the latitude band 78N-82N was used separately to 85N. What distinguishes these regions?
  The text has been modified for clarification:
  Areal-mean time series were constructed using half-day mean values for two distinct areas: the entire inner Arctic pack ice region north of 85° N and the marginal ice zone within the 78° N – 82° N latitude band.

- Lines 163-165: The authors should specify what kind of differential mobility analyzers (and CPCs) were used to measure the aerosol size distributions. What were the size ranges of the DMAs?
  As the present manuscript is a meta-analysis that extends the previous discussions of the particle size distributions of individual expeditions or combinations thereof, we refrained from repeating detailed instrumental characteristics. The references given in the text contained all instrumental details.

- Line 166: What type of "contamination"? Contamination from the ship? Please specify and provide more detail (at least as a brief summary) of how and why this was done.
  The following text has been added to provide more details:
  Direct ship contamination was prevented using a pollution controller connected to the sampling manifold, consisting of a TSI-3025 UCPC linked to the system described by Ogren and Heintzenberg (1990). Additionally, safe wind sectors were identified by releasing smoke from the ship while adjusting wind speed and direction (Leck et al., 1996). If the wind was within ±70° of the bow and above 2 m/s, no ship pollution reached the sample inlets. In 2001 and 2008, a third criterion was added: excluding data when one-minute toluene levels exceeded 75% of their running mean. To maximize pollution-free sampling time, the manifold was kept facing upwind, requiring a "harbor" in the ice to moor the ship and allow rotation with changing wind directions. Details of the respective instrument setups and measures to minimize the risk of contamination from the platform are described in Covert et al. (1996), Leck et al. (2001), and Heintzenberg and Leck (2012).

- Line 168: The authors should specify what kind of aerosol spectrometer was used and what was its size range.
  See our response to lines 163-165

- Line 169-171: For the "harmonized" size distribution interpolation, was any merging done? If so, please describe how and if there was diameter overlap between the different measurements. Please include detail on how considerations were made to combine distributions from mobility and/or optical diameters. An assessment of the merging should also be detailed.
  Sorry about our lack of detail. We added the text: "The size distributions from the five expeditions were harmonized before the fitting procedure by linear interpolation of the measured data on a common set of 50 diameters from 3.37 to 900 nanometers with logarithmically equal spacing. Whenever the measured data did not cover this set completely, the values at the respective interpolation diameters were flagged as 'missing'."
  No consideration was given to combining distributions from mobility and/or optical diameters, as no data from optical instruments were used.

- Line 189: What is meant by the sentence "Vertical layers one through 25 hPa separates five layers"?
  Sorry, the rest of this meaningless sentence was scratched.

- Figure 6 (now Figure 7): The size distributions in these plots show values extending down to 1 nm, while it was described in the methods that the distributions were interpolated to a range of 3.37-900 nm. Can the authors please clarify this discrepancy?
  The lognormal fitting procedure covers the total range of 1-1000 nm.

- Line 301-302: A citation or evidence should be provided here to justify claiming the source of the high nucleation mode as new particle formation in this region. The authors should also clarify what is meant by "new particle formation". I believe the authors mean the formation of particles from gaseous precursors. If so, this should be specified.
  Line 301-302: "In contrast, the size distributions in node 2, also in "melt" indicated strong new particle formation with high nucleation mode concentrations below 10 nm diameter, which will be dealt with below."
  The following text has been added to the above Line 301-302:
  The consideration is limited to new particle formation from the gas phase or from the division of sub-micrometer particles in their airborne state (Baccarini et al., 2020; Covert et al., 1996; Heintzenberg et al., 2006; Karl et al., 2013; Lawler et al., 2021; Leck and Bigg, 2010). A more detailed discussion will follow in section 5.4.

- Line 302: revise "dealt with" to "discussed".
  *We have changed "dealt with" to "discussed".*

- Figure 6 (now Figure 7): What is meant by the y-label in this figure? Are these size distributions showing the number concentration in each bin not the normalized number concentration per dlogDp bin? If it is the former, why is that presentation of the distribution shown rather than the dN/dlogDp (the concentration normalized by the differential log diameter which is more commonly used?
  The y-label of Figure 6 (now Figure 7) has been changed to the more commonly used dN/dlogDp.

- Section 4: Can the authors provide some quantitative statistical information on the modal properties of the size distributions between nodes and melting/freeze-up? Specifically, the authors should quantify the mean diameters and number concentrations of the modes between nodes and freeze-up and melt. How do these properties compare to previous literature if available for synoptic regime differences in the size distribution? What do these results suggest for changes due to aerosol sources, synoptic circulation, and sea ice conditions?
  We added error bars to the graphs in Figure 6 (now Figure 7) to represent the standard deviations of the means of the average size distributions.

- Line 323: Clarify what is meant by "outlier size distributions of node 2 in Fig. 6b." I believe the authors are indicating the high concentration nucleation mode remaining in the melt group. This is true, see below.

- Lines 325-326: Specify that the description "Figure 8c (now Figure 9c) showed an average…" is pertaining to the average of nodes 1,3,4,5, and 6.
  Lines 323; 325-326 are now specified, the text reads:
  The outlier size distributions of node 2 in Fig. 6b, which show a high concentration of nucleation mode remaining in the melt group, can now be understood by referring to the pdfs for "melt" in Fig. 8c, These pdfs represent the average of nodes 1, 3, 4, 5, and 6 and show an overall decrease from a 14% probability of solid sea ice to a low probability of broken ice beyond 50% open water, with a narrow probability extreme of 100% open water.

- Line 326: Revise to "the [sea] ice".
  Now added.

- Lines 372-373: Please clarify what are the "segment ratios".

Explained on line 356.

- Line 379-380: The final sentence in this passage is very confusing and vague. Please clarify/revise or remove.
  The sentence has been removed.

- Lines 410-411: "systematic trends", what is meant by this? Were the trends found to be statistically significant?
  The trends are not statistically significant, but they are obvious, as now stated in the sentence on line 414. This is indirectly followed up on lines 421-422: "Table 2 collects the changes in critical DOY values and length of "melt" and "freeze-up" from 1991 to 2023 for assumed linear developments in the two studied Arctic regions."

- Lines 435-437: In these lines, the differences in the size distributions should be further quantified and compared using statistical information from the modes.
  We added error bars to Figure 12 (now Figure 13), representing standard deviations of the means for the average size distributions and median absolute deviations for the median size distributions.

- Lines 437-439: Can the authors not attribute any of the differences in the size distributions between periods to transport? No evidence in this paper is given to justify claiming these differences are solely driven by new particle formation. A further consideration of these differences needs should be done here. The following has been added to the text for clarity:  The "freeze-up" samples, attributed to Figure 13 (old Figure 12), typically stayed in the air over the pack ice area for over five days before being collected at latitudes of 85° N or higher (e.g., Leck and Svensson, 2015). Extended advection over pack ice has been observed to result in comparatively low particle concentrations for diameters larger than approximately 80 nm, due to scavenging in low clouds and fog, especially during the first 1-2 days of advection from the open sea into the pack ice. (Heintzenberg et al., 2006; Nilsson and Leck, 2002). The tiniest particles, with diameters below 30 nm, have been observed to have very short atmospheric lifetimes, generally ranging from hours to a day (Leck and Bigg, 1999). Therefore, their presence over the inner pack ice cannot be explained by advection from more southerly sources.

- Line 451-470: I believe this entire passage should be significantly reduced to remain within context and on topic of the discussion in this section. As it stands, this passage is a substantial digression. Key findings from the works and

discussion provided here would be more useful to put into context for the proceeding paragraphs.

Lines 447-470 have been replaced with the following: According to Aslam et al. (2016), these changes must influence sea ice distribution, such as open water and newly formed leads. As a result, the response of microorganisms in seawater to melting or freezing ice could have an impact on various biogenic sources at the air-sea interface.

The overall thinning of sea ice, along with earlier and longer melt periods, clearly results in more open water. This increases the sunlight reaching the ocean surface and promotes phytoplankton growth, utilizing the nutrients already available and those supplied by melting ice-bottom algae (Arrigo et al., 2012).

This would also regulate polymer gel production via phytoplankton secretions, as reviewed by Deming and Young (2017) and references therein. However, diminished ice thickness or increased openness of the sea would facilitate more efficient wind mixing of the surface ocean, thereby augmenting the depth of the mixed layer and potentially mitigating algal growth. Because of polymer gels that induce aggregation (Orellana et al., 2011), increased carbon flux from sea ice might occur with earlier ice melt at the MIZ if grazers feeding on aggregates are absent, resulting in less accumulation of polymer gels in the upper water column (Carmack and Wassmann, 2006).

The extension of "freeze-up" by about ten days with freshly frozen leads restricts the exchange of nascent sea spray particles with the atmosphere. However, Bowman and Deming (2010) discovered that frost flowers contain significantly more bacteria and extracellular polymer gels than brines, young ice, or water. Their research indicates that an increase in frost flower occurrence could promote chemical interactions between sea ice and the atmosphere, serving as an enriched atmospheric source of polymer gels as well as for iodine released via the frost flowers from sea-ice brine channels.

Whereas the sum effect of counteracting processes during "melt" on the biogenic Arctic aerosol in a warming climate is unclear, the net impact of the changing "freeze-up" is expected to enhance the biogenic Arctic aerosol in late summer/autumn. In terms of particle size distribution, this may lead to an even more prominent sub-Aitken mode than shown in Fig. 13.

Results from large eddy simulation models indicate that Aitken mode particles could significantly influence the cloud's simulated microphysical and radiative

properties by forming cloud droplets (Bulatovic et al., 2021). Their findings aligned with aerosol particle size distribution data collected during the five-year I/B *Oden* expeditions, which showed that Aitken particles as small as approximately 50-30 nm in diameter can act as CCN. These particles also showed an increased tendency to activate into cloud droplets after the commencement of sea ice formation (Duplesee et al., 2024; Karlsson et al., 2022; Leck and Svenson, 2015). Based on the findings outlined above, the response of microorganisms in seawater to the processes of melting or freezing of ice could significantly impact the formation of low-altitude liquid clouds within the high Arctic environment through aerosol-cloud interactions. This, in turn, may have implications for their radiative properties and the future evolution of the ice cover.

- Lines 527-528: I don't think "aerosol-cloud-climate connections would [require] model simulations". Aircraft, ground-based, and remotely sensed cloud microphysical properties can be retrieved for this purpose by relating them to the observed aerosol measurements of this study and has been done in previous work that should be cited. This should also be mentioned in this concluding discussion.

  There seems to be a misunderstanding of our text. The suggested calculations aim at assessing the possible future effects "a possible doubling of biogenic particle numbers during the freeze-up, and an unknown net source change during the melt season." This cannot be done with existing observations, and we are not aware of previous work in this direction.

  The sentence now reads: The strong aerosol-cloud-climate correlation necessitates regional model simulations to evaluate potential future impacts of a doubling in airborne biogenic particles during the freeze-up period and an indeterminate net source change in the melt season.

The upper panels of Figures 1-4 show the occurrence (%) of each node in each month, and the lower panels show the mean sea level pressure at each node. The colors of the columns in the upper panels correspond to the background colors in the lower panels. Colors also indicate similarities in mean sea level pressure patterns between figures, but they are not precisely the same because each figure is based on an individual SOM analysis. ERA5 data (Hersbach et al., 2025) from 1991 to 2020 for the entire year and for two-month periods.

Figure 1. *Whole year, 12 nodes*

[Figure]

[Figure]

Figure 2. *Whole year 6 nodes*

[Figure]

[Figure]

Figure 3. *Aug – Sep 12 nodes*

[Figure]

[Figure]

Figure 4. *Aug – Sep 6 nodes*

[Figure]

[Figure]

**1. Introduction**

[revised manuscript text omitted]

---

## Author Comment (AC2)

**Referee 1**

**General comment**

The paper investigates potential source regions and ice-related factors affecting size distributions and newly formed particles in Arctic based on measured size distributions during five cruises of I/B Oden in the summers of 1991, 1996, 2001, 2008, and 2018. The topic is of interest and suitable for the Journal. A few aspects should be made more clear in a revision step, see my specific comments.

 We thank referee 1 for a constructive review. Our responses have been incorporated into the revised manuscript. Details are outlined below.

**Specific comments**

Section 2.2, lines 162-167. I understand that details are given in other papers, however, it would be useful to summarise here what instruments are used and their sizer range. In addition, if the different instruments have been compared one with the other and if size distribution of dried (i.e. low RH) air was measured.

All instruments were of the type electrical mobility analyzer. Relative humidities in the instruments were below 20%. Table 1 was complemented with lower and upper size limits pertaining to the different years. The time difference between subsequent expedition years was five to 10 years. Older instruments simply did not exist anymore for any direct comparison.

**Table 1:** Start and end date of hourly I/B *Oden* aerosol data utilized in this study in 1991, 1996, 200, 2008, and 2018, and the number of utilized hourly averaged distributions (Scans) $\geq$ 85° N after screening for possible ship pollution (total 2476). Also shown are the aerosol upper and lower size detection limits of the instruments used in the different expedition years.

| Year | Start date | End date | Lower size limit (nm) | Upper size limit (nm) | Scans |
|---|---|---|---|---|---|
| 1991 | 23 August | 20 September | 3 | 500 | 560 |
| 1996 | 1 August | 9 September | 5 | 600 | 715 |
| 2001 | 1 August | 24 August | 3 | 900 | 503 |
| 2008 | 10 August | 3 September | 3 | 800 | 411 |
| 2018 | 15 August | 16 September | 2.1 | 921 | 287 |

Section 2.3 Do you have any idea of the boundary-layer height. I was wondering if the arrival height was above or within the boundary layer.

The following clarification has been added to Section 2.3: The summer Arctic boundary layer is typically well-mixed and shallow (approximately 100 – 400 m), capped by a temperature inversion. At times, the inversion can be strong, especially when warmer air is advected from lower latitudes while the free troposphere remains stably stratified (Vüllers et al., 2021). An arrival height of 300 meters was chosen to ensure an optimal ensemble configuration, such that the receptor point is within the well-mixed boundary layer and close enough to the aerosol sampling height (25 meters above sea level). Additionally, the chosen receptor height reduces the risk of surface contact in the trajectory calculations caused by rounding errors or interpolation.

Figures 1 and 3 are very small to be readable. Authors should considering using a different layout. Suggestion adopted. Figure 3 (a-f) is now stretched over two figures, labeled Figure 3 (a-c) and Figure 4 (a-c).

Section 4. How the new particle formation events were determined? Or the discussion only consider small particles as newly formed?

The following text has been added to the above Line 301-302: The consideration is limited to new particle formation from the gas phase or from the division of sub-micrometer particles in their airborne state (Baccarini et al., 2020; Covert et al., 1996; Heintzenberg et al., 2006; Karl et al., 2013; Lawler et al., 2021; Leck and Bigg, 2010). A more detailed discussion will follow in section 5.4.

Figure 8 (now Figure 9). What is the meaning of the peak around 100% visible in each graph?

Sorry about our obscure formulation.  The figure caption was extended to "When the probabilities of 0% and 100% open water lie outside the scale of a graph, the respective values are given as numbers in text boxes".

Figure 12 (now Figure 13) and correlated discussion. It seems that there are very large differences between median values and average. This are likely due to large values influencing the average. However, the conclusions obtainable from median would be significantly different compared to those of average. Could you please comment in more detail this aspect. It could be useful to show a comparison of the incidence of new particle formation events in the two cases: melt and freeze-up periods.

In the text on lines 438-439, we write "the averages are strongly affected by individual particle formation events." For more information on general features of small particle formation events, the following text has been added to lines 437-439:

[revised manuscript text omitted]